# Host- plasmid network structure in wastewater is linked to antimicrobial resistance genes

Alice Risely [1,6], Arthur Newbury [2,3,6], Thibault Stalder [4,5], Benno I. Simmons [2], Eva M. Top [4,5], Angus Buckling[2,3] & Dirk Sanders [2,3] ✉

As mobile genetic elements, plasmids are central for our understanding of antimicrobial resistance spread in microbial communities. Plasmids can have varying fitness effects on their host bacteria, which will markedly impact their role as antimicrobial resistance vectors. Using a plasmid population model, we first show that beneficial plasmids interact with a higher number of hosts than costly plasmids when embedded in a community with multiple hosts and plasmids. We then analyse the network of a natural host-plasmid wastewater community from a Hi-C metagenomics dataset. As predicted by the model, we find that antimicrobial resistance encoding plasmids, which are likely to have positive fitness effects on their hosts in wastewater, interact with more bacterial taxa than non-antimicrobial resistance plasmids and are disproportionately important for connecting the entire network compared to non-antimicrobial resistance plasmids. This highlights the role of antimicrobials in restructuring host-plasmid networks by increasing the benefits of antimicrobial resistance carrying plasmids, which can have consequences for the spread of antimicrobial resistance genes through microbial networks. Furthermore, that antimicrobial resistance encoding plasmids are associated with a broader range of hosts implies that they will be more robust to turnover of bacterial strains.

Plasmids play a key role in the spread of antimicrobial resistance (AMR) and other genes (e.g., metal resistance, biodegradation, virulence), both within and between bacterial taxa[1–5]. Understanding the ecological mechanisms that underpin plasmid transmission within bacterial communities is important for combating the spread of AMR and associated bacterial epidemics[6]. However, our knowledge about how plasmids interact with their hosts, which host they can infect, and the dynamics of spread within communities is mostly gained from laboratory research on a limited number of bacteria and plasmids. Therefore, there remains considerable uncertainty surrounding the role of plasmids within larger communities in nature. This limits our understanding of the ecological and evolutionary processes driving plasmid transmission across natural microbial communities. An ecological network approach, where plasmids are linked to their bacterial hosts in natural environments, can provide important insights about the realised plasmid host range (observed range within a specific community) and AMR transmission pathways across these networks.

While natural plasmid–host networks are likely to be very complex and affected by a wide range of variables such as nutrients and habitat structure[7,8], it is possible to make some general predictions.

[1]School of Science, Engineering, and Environment, University of Salford, Salford M5 4WT, UK. [2]Centre for Ecology & Conservation, College of Life and Environmental Sciences, University of Exeter, Penryn, Cornwall TR10 9FE, UK. [3]Environment and Sustainability Institute, University of Exeter, Penryn, Cornwall TR10 9FE, UK. [4]Department of Biological Sciences, University of Idaho, Moscow, ID, USA. [5]Institute for Interdisciplinary Data Sciences, University of Idaho, Moscow, ID, USA. [6]These authors contributed equally: Alice Risely, Arthur Newbury. ✉e-mail: d.sanders@exeter.ac.uk

Theoretical and experimental evidence suggests that plasmids that have positive fitness consequences for their hosts should interact with a wider range of hosts[9], thereby being central for the formation of network structure. Classical single host–single plasmid models demonstrate that persistence, as well as the frequency of a plasmid in a host population, are largely determined by the fitness effects it has on its hosts[10], with fixation in the population expected when the plasmid confers a benefit[9]. Furthermore, a plasmid within a microbial community need only be able to maintain a high frequency in one host population to continually re-infect other bacteria in the community[11]. Thus, even if a plasmid is only beneficial to one or a subset of bacterial strains in a community, it may maintain a relatively high frequency amongst other strains[9]. Such exposure to and conjugation between multiple bacterial strains can also lead to the evolution of generalism in plasmids[12], i.e. a reduction in the costs they impose on multiple hosts. Therefore, beneficial plasmids (at least to some hosts) should have higher contact rates with a range of hosts, potentially leading to the evolution of generalist plasmids. So far, however, there is (1) a lack of theory relating to host–plasmid network formation in the case of multiple (potentially incompatible) plasmids. This is a significant limitation since natural communities contain a wide range of plasmids. Moreover, there is (2) an absence of real-world data, comparing the structure of beneficial versus non-beneficial plasmid–host networks, which in the presence of antibiotics equates on average to plasmids that carry relevant AMR genes versus plasmids that do not.

In this work, we combine theory and observation to address both issues. We use a plasmid population model to explore the relationship between the positive vs negative effects plasmids have on their hosts and the number of hosts they interact significantly with when embedded in a community with multiple hosts and multiple plasmids. Recent work applied a similar model to investigate the persistence of a single plasmid in a microbial community[13]. We extend this approach to include multiple plasmids, whose fitness effects vary and are unique to the specific host–plasmid combination. This addition to the modelling framework is crucial for making predictions about plasmid population dynamics in natural communities. To construct and analyse a host–plasmid interaction network based on a natural microbial community we make use of recent technical and analytical advances. We use Hi-C metagenomics[14–16] to link mobile genetic elements to their

bacterial hosts in a wastewater sample[14]. We determine host–plasmid linkage from this natural microbial sample by using geNomad[17], a recently developed identification tool for mobile genetic elements, to identify clusters of plasmid contigs that originate from the same cell (hereafter termed 'putative plasmids'). We test whether AMR presence on these putative plasmids is associated with altered interaction distributions with 374 bacterial metagenome-assembled genomes (MAGs). While plasmids can carry a range of host-beneficial genes, we make the assumption that in municipal wastewater putative plasmids that carry AMR genes will be on average more beneficial than those that do not. Antibiotics are commonly found in wastewater[18–22] often exceeding minimal inhibitory concentrations (MICs) and predicted no-effect concentrations (PNECs), especially before treatment[23,24]. Therefore, antibiotics in wastewater are likely to confer an advantage to the majority of AMR genes[25,26]. Here, we show that plasmids that carry AMR genes interact with more bacterial hosts (i.e., have a higher degree in the network, which describes the number of connections the plasmid has to other host taxa) than plasmids without AMR genes, and thereby lead to a more connected network. We further discuss the potential of this analytical approach to understand the spread of AMR.

## Results
### Model
Our population model extends the work by Alonso del Valle et al.[13] and describes the dynamics of three plasmids in a community of eight bacterial strains. Plasmids have varying fitness effects ranging from imposing fitness costs to being beneficial. For each community, once the model reaches equilibrium, we calculated the network degree for a focal plasmid as the number of associated bacterial hosts in which it makes up at least 1% of the population.

Plasmids that benefited a subset of hosts in the model showed a higher network degree and this degree increased with the number of hosts they benefit. However, there was a non-linear relationship between positive interactions, conjugation rate and degree, whereby the increase in degree caused by positive interactions was greatest at an intermediate conjugation rate value $10^{-7}$ (Figs. 1a and S1)). Here, conjugation rates were the same for each plasmid/host combination in each simulation run, but with a 10-fold increase in conjugation rate within vs between host strains. When conjugation rates were lower,

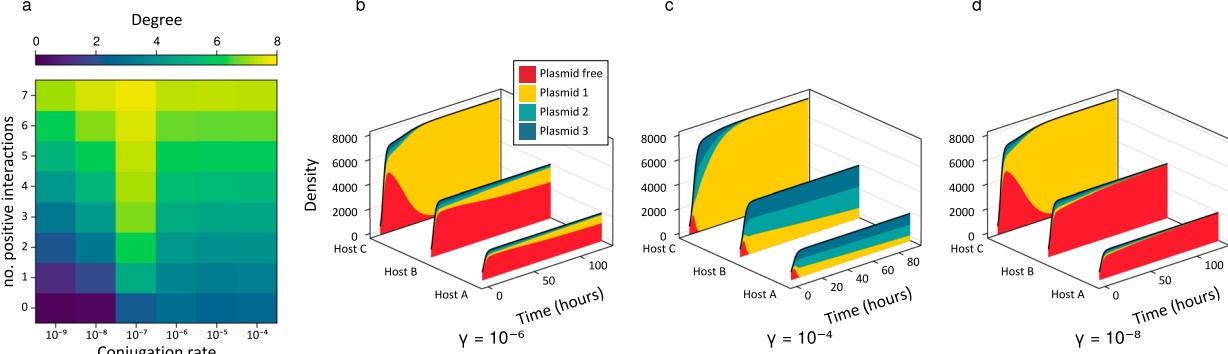

**Fig. 1 | Plasmid population dynamics model results. a** Mean degree (number of hosts with at least 1% infection rate for the focal plasmid) of a plasmid embedded in a community of eight hosts and three plasmids. Estimates are shown for a range of conjugation rates and the number of positive interactions for the focal plasmid, whilst both other plasmids confer only negative effects on hosts and all other parameter values are assigned randomly as detailed in the "Methods" section. Panels **b**–**d** highlight the way in which a plasmid which is beneficial to a subset of hosts can spread throughout the rest of the host community as a result, and show why the results in **a** are sensitive to conjugation rate $\gamma$. Each panel shows the first 100 h of a 3 × 3 bacteria plasmid network. Here, all three plasmids have identical negative interactions with all three hosts, apart from plasmid 1, which has a positive

interaction with host C and is slightly more costly to hosts A and B than the other two plasmids. **b** As this plasmid spreads through the host population, it increases host C's density and resultantly transfers to the other two hosts at a high rate, supplanting the other two plasmids. In panel **c** with a 100-fold increase in conjugation rate, all three hosts quickly become approximately fully infected. Since conjugation frequency then becomes negligible with so few susceptible bacteria, fitness effects determine which plasmids will thrive in a given host population. Thus, plasmid 1 is steadily being lost from hosts A and B. In panel **d** there is a 100-fold decrease in conjugation rate compared with (**b**). Here, all three plasmids are being lost from hosts A and B since the rate of conjugation is not sufficient to overcome the fitness costs incurred.

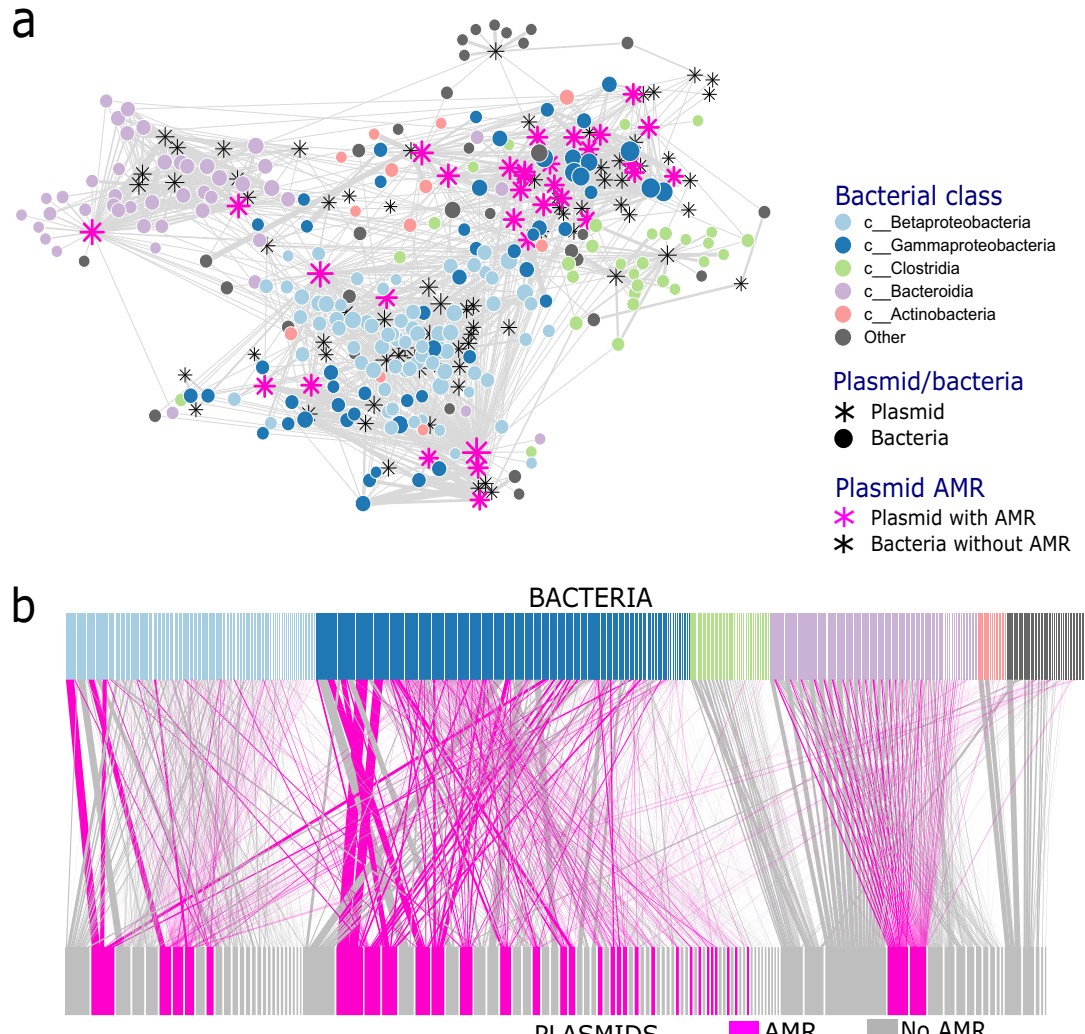

**Fig. 2 | Bacteria–plasmid networks based on normalised Hi-C linkage. a** The full bacteria–plasmid network is made up of 374 bacterial MAGs and 109 putative plasmids, with MAGs coloured by class. Plasmids are represented by stars, and plasmids with AMR are highlighted in pink. Node size represents how many other nodes they are connected to; **b** same network as (**a**) but visualised as a bipartite network, with plasmids carrying AMR genes and their links to MAGs highlighted in pink. The size of the bars represents how many Hi-C connections the MAG or plasmid had. The width of the edges in both **a** and **b** represents the strength of the association (i.e., thicker edges represent more Hi-C connections). Data to replicate this figure is contained in the source data file.

there was less opportunity for plasmids prevalent within a particular host to invade another host population via inter-strain conjugation. Whereas, at higher conjugation rates, the number of susceptible bacteria within each population fell more rapidly, leaving less opportunity for plasmids to move between strains and allowing the within-host dynamics to be governed more by the fitness effects of each plasmid (Fig. 1b–d). The same pattern emerges for a range of growth rates and plasmid loss parameters (shown in Figs. S1 and S2). Overall, as with previous models involving only a single plasmid[9], beneficial interactions lead to a higher degree in the network, though here with multiple incompatible plasmids degree did not increase linearly with the conjugation rate.

### Wastewater network

We tested whether the predictions from the population model were empirically supported by a bacteria–plasmid network generated from a natural microbial community. The inferred wastewater bacteria–plasmid network was made up of 374 bacterial MAGs (metagenome-assembled genomes) and 109 putative plasmids (Fig. 2a, b). Most MAGs identified belonged to either the class Betaproteobacteria (Phylum Pseudomonadota), Gammaproteobacteria (Phylum

Pseudomonadota), Clostridia (Phylum Firmicutes), Bacteroidia (Phylum Bacteroidetes) or Actinobacteria (Phylum Actinobacteria). The full network clustered strongly by bacterial taxonomy, with MAGs belonging to Pseudomonadota, Firmicutes, and Bacteroidetes largely clustering separately (Fig. 2a). The large majority of AMR plasmids (21 of 32) clustered together with Gammaproteobacteria (Fig. 2a, b), with this Gammaproteobacteria cluster heavily represented by the genus Acinetobacter. Most of the AMR genes found on Gammaproteobacteria-associated plasmids were identified as the protein msr(E) (Fig. S3), an ABC-F subfamily protein that confers resistance to a range of antibiotics including erythromycin and streptogramin[27]. Taxa belonging to Bacteroidia were associated with two plasmids carrying tet(Q) and aadS proteins, associated with tetracycline and streptomycin resistance, respectively[28,29].

Bacterial MAGs were associated with an average of 4 putative plasmids (median = 2, min. = 0, max. = 42), whilst putative plasmids were associated with an average of 14 MAGs (median = 10, min. = 1, max. = 112). Note that this does not equate to one bacterial cell having 42 plasmids; rather, 42 putative plasmids were found to be associated with that MAG across its entire population within the wastewater community, which describes the degree of that MAG in the network.

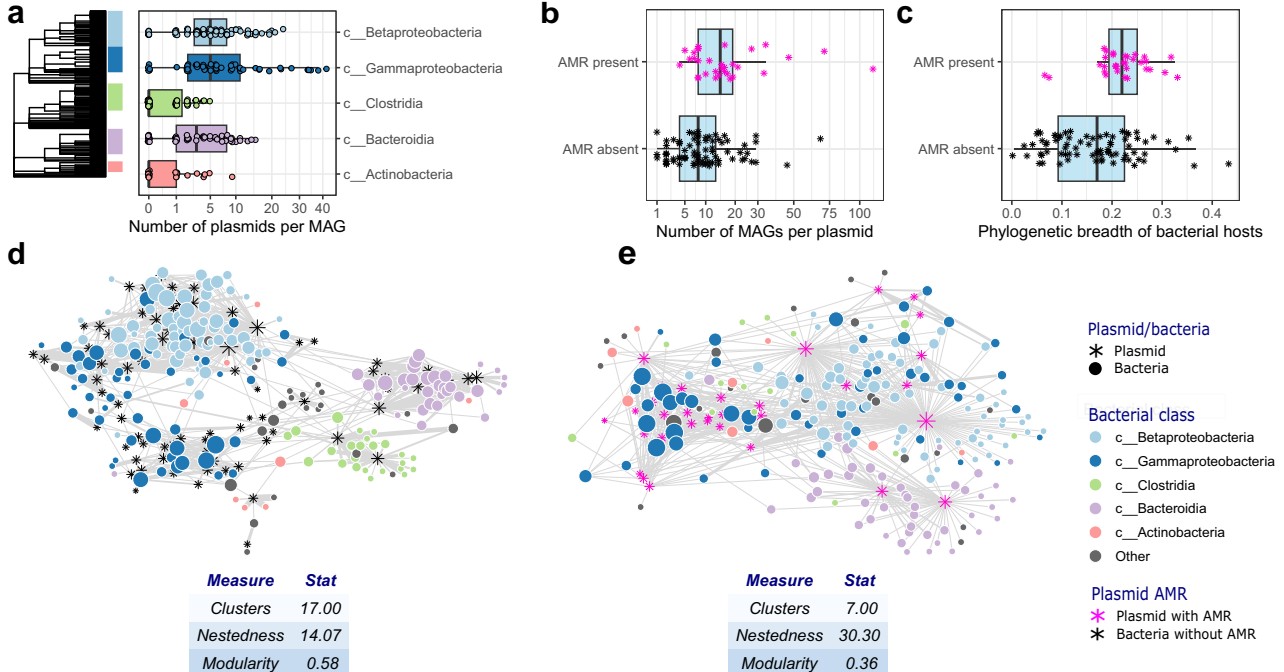

**Fig. 3 | Network structure and effect of AMR plasmids, based on 374 bacterial MAGs and 109 putative plasmids. a** Number of plasmids per bacterial MAG, split by bacterial class. Points represent unique MAGs. **b** Number of MAGs per putative plasmid, split by whether plasmids were associated with antimicrobial resistance genes (AMR). **c** Phylogenetic breadth of bacterial hosts, split by whether plasmids were associated with AMR. Points in **b** and **c** represent unique putative plasmids; Sub-networks and network statistics when retaining only **d** plasmids without AMR genes and **e** plasmids with AMR genes. Stars represent plasmids and circles MAGs, with AMR plasmids highlighted in pink. Widths of the edges represent the strength of the Hi-C connection, whilst node size reflects node degree. The phylogenetic tree in **a** is based on bacterial taxonomy. Boxplots for **a**–**c** show the interquartile range and median. The whiskers extend no further than 1.5*IQR from the hinge. Data to replicate this figure is contained in the source data file.

MAGs that associated with a high number of plasmids were distributed across the phylogenetic tree, although MAGs belonging to Betaproteobacteria, Gammaproteobacteria, and Bacteroidia tended to associate with a higher number of plasmids than those belonging to Clostridia and Actinobacteria (Kruskal $X^2 = 88.7$, $p < 0.001$; Fig. 3a). Putative plasmids that were associated with AMR genes were more widely distributed across MAGs (median AMR = 15.5, median no AMR = 8; Wilcoxon $W = 750$, $p = 0.0013$; Fig. 3b) and more phylogenetically diverse suite of bacterial hosts (Wilcoxon $W = 677$, $p = 0.0002$; Fig. 3c) than those that were not associated with AMR genes.

We investigated the link between likely beneficial plasmids and network structure by comparing sub-networks based on whether putative plasmids are associated with AMR or not. When putative AMR plasmids were excluded (Fig. 3d; 205 bacterial MAGs and 32 plasmids), networks were made up of more clusters, were more modular (have a higher number of separated sub-networks that clustered strongly by class) and less nested (nestedness means the presence of highly generalist plasmids linking the network, with rare specialist plasmid links are already provided by the generalists) than networks based on solely AMR putative plasmids (Fig. 3e; 203 bacterial MAGs and 77 plasmids). Putative plasmids carrying AMR genes further connected Pseudomonadota to other phyla linking large parts of the whole network. We additionally focused on networks formed by Pseudomonadota and their plasmids and found similar differences in network structure within this bacterial phylum compared to the full network (Fig. S4).

We next visualised the distribution of the 10 putative plasmids with the highest network degree (super generalists) across the bacterial phylogenetic tree (Fig. 4). Putative plasmids with the highest degree were largely shared amongst members of the same phylum, although some were occasionally shared more widely. Putative plasmids that were associated with Pseudomonadota were often shared across both Beta- and Gammaproteobacteria (Fig. 4). Whilst most putative plasmids remain undescribed, plasmid 2 (Fig. 4) was identified as a broad range plasmid belonging to the IncP-β group.

## Discussion

Our model predicts that in communities with multiple bacterial strains and plasmids, beneficial plasmids will be associated with a higher number of unique hosts than costly plasmids. The network analysis of a natural complex microbial community from a wastewater sample reflects those predictions, with both approaches providing critical insights into the ecology of AMR spread by plasmids in complex communities. We found that while the natural plasmid–host community in the wastewater sample was dominated by more specialist putative plasmids, those carrying AMR genes tended to be more generalist and increased the connectivity of the overall network. As predicted by the model, the network structure for the AMR plasmid–host subnetwork differed from the non-AMR plasmid network. The AMR plasmid–host network showed higher plasmid generalism, with an overall higher level of connectedness. Further, we found much clearer phylogenetic structuring in the non-AMR plasmid network. This shows that AMR plasmids play an important role in connecting MAGs across bacterial phyla.

Our model and data from experimental microbial communities[9,30] suggest that the reason for these patterns are two different broadly categorised types of plasmids: (1) Those that are costly for the host are more specialised and (2) plasmids that are beneficial for the host are more generalist. In our model, beneficial plasmids reached higher prevalence in their hosts which then increased the likelihood of transmission to other strains. These dynamics lead to highly connected plasmid–host networks. It is reasonable to think that AMR genes can be directly beneficial in the wastewater environment

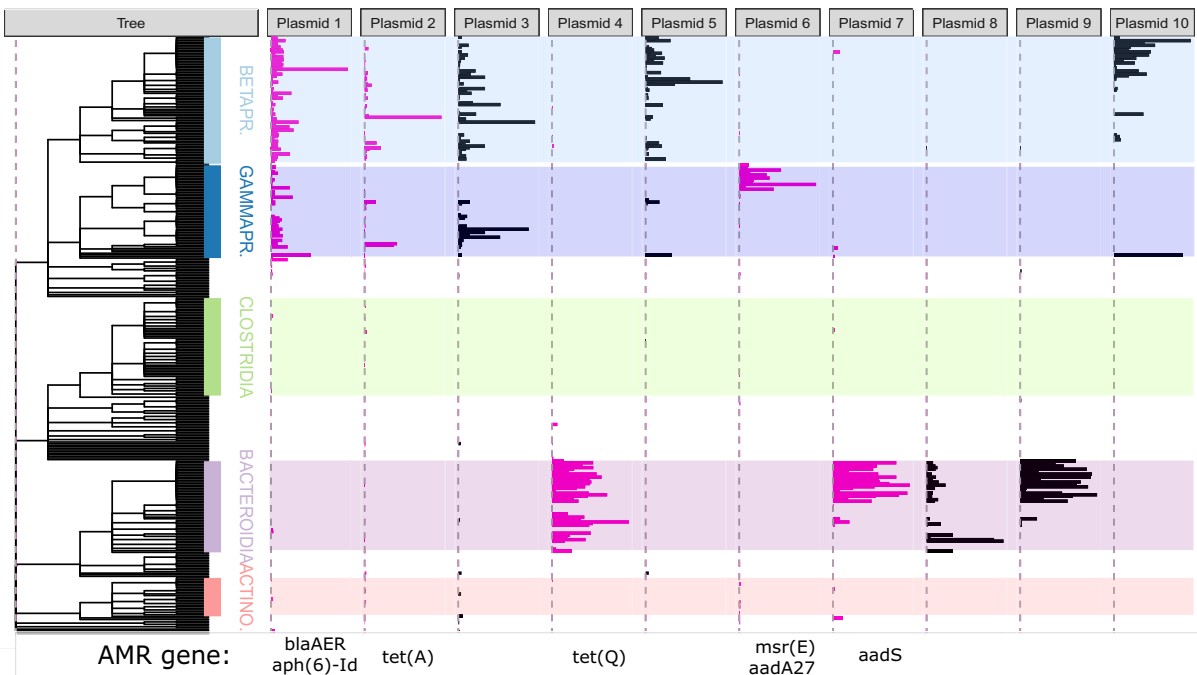

**Fig. 4 | Phylogenetic distributions of the most prevalent plasmids across 374 MAGs.** Phylogenetic weighted distributions of the 10 most prevalent putative plasmids (i.e. plasmids with associations to the highest number of MAGs), ordered by how many MAGs they associate with. Bar length represents interaction strength (i.e. the number of normalised Hi-C links). Putative plasmids with AMR are highlighted in pink, and the AMR-associated gene or genes are indicated beneath. The phylogenetic tree is based on taxonomy. Data to replicate this figure is contained in the source data file.

because they can confer a selective advantage even in the presence of low concentrations of antibiotics and other biocides[31]. Even if not directly beneficial in the current wastewater environment experienced by these organisms, AMR genes will almost certainly have provided a benefit in the environments they originate from, such as hospitals and a community of people consuming various antibiotics. In the presence of antibiotics, we can expect the dynamics predicted by the model, with beneficial AMR plasmids becoming more prevalent, spreading faster and coming in contact with a higher number of bacteria. Recent experiments using simplified bacteria–plasmid networks demonstrate that short-term growth rate advantages conferred by a beneficial plasmid can result in greater plasmid generalism[9]. Specifically, if a plasmid increases the frequency of its host, the plasmid then has greater opportunities to be transmitted to other host taxa. While the distribution of hosts of a plasmid can be influenced by factors affecting its ability to transfer into a new host, after entry it is primarily the plasmid-encoded replication system and its interaction with host factors that determines the ability of a plasmid to survive in that host[32,33]. This suggests that increased ecological generalism of beneficial plasmids could in turn promote greater evolutionary generalism as a consequence of mutualistic coevolution for plasmid maintenance occurring between plasmids and the multiple hosts they interact with[34]. Note that we can't distinguish ecological and evolutionary generalism within the natural community studied here.

Interestingly, observational studies across biological systems such as plant–insect or tree–parrot communities describe similar patterns, where interaction networks dominated by (trophic) antagonists tend to have fewer generalist interactions than mutualistic networks[35–37]. Theory suggests that more generalism may lead to greater stability in communities dominated by mutualistic interactions (i.e., species and functional traits are less likely to go extinct, resulting in changes in network properties), while generalism decreases the stability of antagonistic communities[35]. Theory also predicts that coevolution may drive this pattern under the assumption that trait matching (e.g., attack-defence traits) determines the strength of antagonistic interactions while trait differences (e.g., barriers for transmission) determine mutualistic interactions[38,39].

While greater generalism associated with AMR plasmids obviously has important implications for the spread of the specific AMR genes encoded by the plasmids, it is also likely to affect the spread of additional AMR genes, even those not currently under selection. First, a new AMR gene that gets incorporated into a generalist plasmid will have more chance to spread. Second, generalist plasmids are more likely to acquire additional AMR genes (e.g., by transposition), given the greater diversity of hosts they interact with. However, it is likely that there are higher costs for the hosts associated with a higher number of AMR genes on mobile genetic elements, suggesting that there is an upper limit[40]. Generalist plasmids and in particular generalist plasmids with AMR were mostly associated with Gammaproteobacteria and the genus Acinetobacter, supporting other findings on the importance of this genus for harbouring diverse AMR profiles[41]. Generalist AMR plasmids assumed a central role in the overall network by linking Pseudomonadota to other phyla, although these interactions were relatively rare. These generalist plasmids may contribute to AMR gene transfer in general between different phyla of bacteria. This might reflect a greater selection for AMR in Pseudomonadota because many common human pathogens belong to this phylum. Two meta-analyses observed that the accumulation of AMR resulting from mutations of chromosomal genes entails a much stronger fitness cost than the accumulation of transferable AMR genes from plasmids[40,42]. This phenomenon may contribute to the observed dominance of transferable AMR genes in the current multidrug resistance epidemic in enterobacteria.

Our approach to analyse the wastewater community advances on the analysis of Stalder et al. [14] by utilising previously validated methods (geNomad[17]) to identify undescribed plasmid signatures. Whilst this method considerably increases our understanding of how undescribed plasmids contribute to interaction network structure, we assume that

connected clusters of sequences represent one plasmid. Yet, it is possible that these clusters in fact represent multiple co-occurring plasmids, or, conversely, that some sequences treated as separate plasmids are in fact part of the same plasmid. An additional limitation to our approach is that some shared genes or mobile elements between different plasmids could have amplified the connections of generalist putative plasmids to more hosts. We strived to remove any such genomic elements, such as transposons, AMR, metal resistance, biocide resistance and virulence genes, yet it is possible that at least some generalist putative plasmids may be a product of other plasmid accessory genes commonly shared among different plasmids of this bacterial community.

Future advances in Hi-C technology paired with long-read sequencing methods will further our ability to distinguish and describe plasmids in natural communities using high-throughput sequencing technology. Analysing the network structure of the wastewater sample is based on a correlational study and AMR presence and generalism may also be driven by host taxa. Indeed, Pseudomonadota is a very ecologically diverse phylum[43], so they may be more likely to be associated with promiscuous plasmids that carry genes beneficial in a range of hostile environments. However, the analysis of the Pseudomonadota network for AMR and non-AMR plasmids showed similar results compared to the overall analysis (see Fig. S4). To fully understand the ecological and evolutionary dynamics under varying plasmid–host interaction types we need experimental approaches that measure fitness consequences and link those to changes in observed network structures.

The modelling approach employed here is more complicated than most plasmid population models, due to the incorporation of multiple hosts, plasmids and resources. However, some simplifying assumptions were necessary. Firstly, we assume bacterial hosts interact only through competition for resources. While in reality bacteria interact via diverse mechanisms such as the excretion of both nutrients and toxins, such features are unconnected to the core focus of the present works—the increased spread of plasmids throughout a host community due to benefits conferred within populations. That said, the impact of host community structure (as a result of inter and intraspecific interactions) on host–plasmid networks is itself an interesting question for future research. The model was also simplified by considering multiple plasmids from within the same incompatibility group. This avoids making further assumptions about how the fitness effects of multiple plasmids interact in natural communities. Furthermore, it has been shown previously that a plasmid that is not competing with other plasmids for hosts will form a better-connected plasmid–host network when it is beneficial[9]. Thus, it is not expected that considering multiple plasmids which are compatible with each other would change our overall result. Though again, the exact effects of the distribution of incompatibility groups on plasmid–hosts networks are itself a useful line of enquiry for future research.

By conducting ecological network analyses on a wastewater Hi-C metagenome, we have been able to describe a natural plasmid–host network. The patterns we observe are consistent with the predictions made by our model. First, the network is primarily driven by specialism, consistent with a predominantly parasitic impact of plasmids in the absence of carriage of beneficial accessory genes. Second, a greater prevalence of AMR genes—which are often transferred by plasmids—in generalist and abundant plasmids leads to a more connected network. Third, the sharing of a few generalist plasmids across the network promotes the potential for inter-class HGT and indirect network interactions. A fourth implication of our results is the resilience of AMR plasmids to ecological processes affecting bacteria. Since AMR plasmids are typically associated with a range of hosts, their presence is less vulnerable to the local extinction of one or a few host taxa. Further work is clearly required to determine the generality of our findings and the mechanisms underpinning them. This also

includes other types of networks, such as bacteria–bacteriophage[44], where interactions while primarily antagonistic can also be mutualistic[45,46]. A closer look at the types of plasmids that cause higher network connectedness would also help understand the drivers of AMR spread in various environments.

## Methods
### Model
To understand how plasmid fitness consequences impact the formation of host–plasmid networks and affect plasmid spread in a community we used a population modelling approach. This approach is based on Eqs. (7)–(9) from Alonso-del Valle et al.[13], extended to accommodate multiple plasmids and multiple resources, which are part of a bacterial community of $M$ strains, initially harbouring $N$ plasmids and subsisting in $L$ resources. To make predictions about long-term community dynamics we extended the model in three ways: (i) resources accumulate according to a logistic function

$$\hat{R}^k R^k \left(1 - R^k\right) \tag{1}$$

where $R^k$ is the $k$th resource and $\hat{R}^k$ its associated rate of influx. (ii) Resource depletion due to consumption is proportional to the densities of the consumers (bacteria with and without plasmids)

$$\sum_{i=1}^{M} U_0^{ik}\left(R^k\right) B_0^i + \sum_{i=1}^{M} \sum_{j=1}^{N} U_p^{ijk}\left(R^k\right) B_p^{ij} \tag{2}$$

where $B_0^i$ is the density of cells the $i$th bacterial strain without any plasmid and $U_0^{ik}$ is the function determining the rate at which it consumes the $k$th resource. Likewise, $B_p^{ij}$ is the density of the $i$th strain with the $j$th plasmid and $U_p^{ijk}$ its consumption function for the $k$th resource.

$$U_0^{ik}\left(R^k\right) = \frac{R^k + V \max^{ik}}{Km + R^k} \tag{3}$$

where $V \max^{ik}$ is the maximum consumption rate of the $k$th resource by the $i$th strain and $Km$ is a half-saturation constant, common to all strains and resources. (iii) We allow for cell death in the model at a rate $d^i$ for each strain, which increases linearly with the total density of the relative strain (with and without plasmids) ($B_0^i + \sum_{k=1}^{M} B_p^{ik}$).

As with Alonso-del Valle et al.[13], bacterial reproduction is given by the function $G(R) = \rho U(R)$, where $\rho$ is the strain-specific efficiency of converting consumed resources into new cells. Thus, given $G_0^{ik}$ is the function governing the conversion of the $k$th resource into cells of the $i$th species (plasmid free) and $G_p^{ijk}$ is the same for the $i$th species with the $j$th plasmid, the dynamics of the resources, plasmid-carrying and plasmid-free cells are governed by the following differential equations:

$$\frac{dR^k}{dt} = \hat{R}^k R^k \left(1 - R^k\right) - \sum_{i=1}^{M} U_0^{ik}\left(R^k\right) B_0^i - \sum_{i=1}^{M} \sum_{j=1}^{N} U_p^{ijk}\left(R^k\right) B_p^{ij} 1 \tag{4}$$

$$\frac{dB_p^{ij}}{dt} = \left(1 - \lambda^i\right) \sum_{k=1}^{L} G_p^{ijk}\left(R^k\right) B_p^{ij} + \sum_{k=1}^{M} \delta^{ik} \gamma B_p^{kj} B_0^i - d^i B_p^{ij}\left(B_0^i + \sum_{k=1}^{M} B_p^{ik}\right) \tag{5}$$

$$\frac{dB_0^i}{dt} = \sum_{k=1}^{L} G_0^{ik}\left(R^k\right) B_0^i + \lambda^i \sum_{j=1}^{N} \sum_{k=1}^{L} G_p^{ijk}\left(R^k\right) B_p^{ij}$$
$$- \sum_{k=1}^{N} \sum_{j=1}^{M} \delta^{ij} \gamma B_p^{jk} B_0^i - d^i B_0^i\left(B_0^i + \sum_{k=1}^{M} B_p^{ik}\right) \tag{6}$$

where $\lambda^i$ is the $i$th strain's rate of loss of plasmids due to segregation, $\gamma$ is the conjugation rate (shared by all hosts/plasmids) and $\delta$ is a square matrix in which all off-diagonal elements equal 1 and all diagonal elements equal 10 resulting in 10-fold higher intra-strain that between-strain conjugation rate. Note that since conjugation is a function of the contact between plasmid-carrying cells (for a given plasmid) and plasmid free cells of a potential recipient, we are assuming no coinfection of distinct plasmids in a single host cell, i.e., plasmids belong to the same Inc group. This simplifying assumption greatly reduces the complexity of the model, while keeping the focus on the key distinction between single and multiple plasmid models (the ability for the spread of one plasmid to be affected by another). Furthermore, it avoids making additional assumptions about how plasmid fitness effects interact.

In order to measure the impact of positive interactions on the network degree of plasmids (their number of hosts) we numerically solved Eqs. (1)–(3) with $L = 3$, $N = 3$ and $M = 8$ for 133,480 unique combinations of parameter values, based on the empirical parameters estimated in Alonso-del Valle et al.[13] using the fifth-order explicit Runge–Kutta method[47] implemented in DifferentialEquations.jl[48] in the Julia programming language[49]. For the results presented in the main text, we set $\lambda^i = 10^{-6}$ for each $i$, $Km = 1$ with $\rho^i \sim$ truncated-Normal ($8 \times 10^8, 100, 4.8 \times 10^8, 1.2 \times 10^9$) and $V\max^{ik} \sim$ truncated-Normal ($6Km \times 10^{-10}, 100$), $4Km \times 10^{-10}, 8Km \times 10^{-10}$), though qualitatively similar results were obtained for a range of other parameter values (Figs. S1, S2). To produce a range of predominantly negative fitness effects, fitness effects $w^{i \cdot j}$ for each strain$^i$ and plasmid$^j$ combination were drawn from $N(0.985, 0.007)$ with the resulting value used as a multiplier: $\rho^{ij} = \rho^i w^{ij}$, $V\max^{ikj} = V\max^{ik} w^{ij}$. Thus, for $w < 1$, the plasmid has a negative effect. We truncated the distribution of $w$s at 1 for 2 of the three plasmids, leaving only a single (focal) potentially beneficial plasmid in each community (see Fig. S2 for an alternative approach, where all three plasmids may be beneficial within a single simulation). Once equilibrium was reached, we calculated the network degree for the focal plasmid as the number of bacterial hosts in which it makes up at least 1% of the population. Mean degree values for a range of conjugation rates and numbers of positive interactions were visualised using Makie.jl[50].

## Hi-C dataset/sample origin
The Hi-C metagenome data from Stalder et al.[14] are derived from a wastewater sample corresponding to a 24-flow composite sample (repeated sampling over a 24 h period and combined) collected in 2017 at the Moscow wastewater treatment plant in Idaho (USA). The facility serves a population of ~25,000 residents and collects mainly domestic wastewater. Hi-C metagenome data uses proximity ligation technology to link contigs that are in close physical proximity (i.e., in the same cell) and therefore can be applied to improve genome assembly and link bacterial hosts to mobile elements such as plasmids.

## Sample processing
Full details of how the wastewater sample was processed are described in Stalder et al.[14]. In brief, half of the homogenised sample was used to generate Hi-C libraries using the ProxiMeta™ Hi-C preparation kit (Phase Genomics, Seattle, WA, USA) and the other half was used to generate the shotgun library. For the shotgun library, total genomic DNA was extracted and isolated using the DNeasy® PowerWater® kit (Qiagen, Venlo, Netherlands). PCR-free Illumina libraries for short insert length sequencing with Hiseq were made by the IIDS Genomics Resources Core (Moscow, ID, USA) using TruSeq® DNA PCR-Free library Prep kit (Illumina, San Diego, CA, USA). Hi-C and shotgun metagenomic libraries we pooled and sequenced using HiSeq 4000, $2 \times 150$ bp paired-end reads at the University of Oregon sequencing core (Eugene, OR, USA).

## Hi-C data processing
The major Hi-C processing steps are visualised in a flowchart (Fig. S5). To build our host–plasmid network, we categorised every metagenome contig as belonging to either a bacterial MAG or a plasmid, and categorised plasmids by whether they were connected via Hi-C to an AMR gene or not. Any contigs that did not identify as either of these elements were automatically removed from the dataset. Contigs that were identified as potential transposons were removed from the dataset, as their ubiquity can generate spurious associations[51]. Transposons and IS elements were identified by performing a homology search with BLASTp on predicted genes from all contigs using an $e$-value < 0.01 against all known transposase proteins from the databases from IS finder[52] available from https://github.com/thanhleviet/ISfinder-sequences/blob/master/README.md and from Tn3 Transposon Finder[53] available from https://tncentral.proteininformationresource.org/TnFinder.html. Protein-coding genes were predicted from all contigs using prodigal in metagenomic mode using the option '-p meta' available from https://github.com/hyattpd/Prodigal[54]. We then used the Hi-C linkage data to identify how these contigs were attached to one another and to build an interaction network between plasmids and MAGs. Below we summarise each step in this process.

## Generating bacterial MAG data
Hi-C metagenome data was assembled into MAGs using an updated algorithm of ProxiMeta™ on April 4, 2021 (Phase Genomics, Inc. 2021). This generated the 374 MAGs analysed in this study. We assigned MAG taxonomy by running MAGs through Phylophlan[55], which calls MASH[56] for taxonomic assignment, and used taxonomy as a proxy for phylogeny (Supplementary Data 1). While MAGs relate to a metagenomic bin that can be asserted to be a close representation of an actual individual genome, here the quality of most of the MAGs identified (according to genome size and completeness scores computed by CheckM) did not allow us to make this assumption. Instead, we considered MAGs not as one bacterial cell's genome, but as the genome (or fragments of a genome) of closely related strains within a species. MAG abundance was calculated as the percent of the average Hi-C read depth of the assembly contained in the MAG relative to the average Hi-C read depth of the total assembly of the sample.

## AMR gene detection
Contigs with genes coding for antimicrobial resistance, virulence factors, metal resistance or resistance to biocides were identified using AMR finder plus[57], using '-n' and '--plus' parameters. The vast majority of resistance genes were categorised as AMR, therefore we only considered genes as 'AMR' if they coded for antimicrobial resistance and not metal resistance or virulence factors. AMR contigs were used to identify which putative plasmids carried AMR genes but were subsequently removed from the network because, similar to transposons, their ubiquity can create spurious network links.

## Identifying putative plasmids
To identify contigs of plasmid origin, all contigs were run through geNomad[17] using the end-to-end command. Because the contigs identified as plasmids were mostly constituted by short contigs (the median length was 3703 bp), we reasoned that for a contig to belong to a plasmid it should be consistently connected to at least one other such contig. To account for this, we retained only plasmid contigs that were linked to other plasmid contigs at least 15 times ($n = 379$ contigs). This may exclude small non-transferable plasmids, but not conjugative or mobilisable plasmids[58] that are the focus of this study. We then performed a cluster analysis on these plasmid contigs using the Walktrap clustering algorithm using the igraph::walktrap.community function[59] and with a step length of 10. This clustering step identified

109 plasmid clusters which we treated as putative plasmids. We did not find any matches with the described plasmids identified by PlasmidFinder[60]. However, we manually checked gene content of the top ten putative plasmids and detected transfer genes belonging to the IncP-β group on the second most prevalent plasmid.

We conducted several quality checks to assess the reliability of these clusters of contigs we called putative plasmids. We first checked the total length of plasmid contigs. The average total length (19,368 bp) and the general distribution (median = 8772 bp, min = 1351 bp, max. = 240,077 bp) are within the expected range of plasmid sizes found in natural communities[61]. Associations characterised by less than five Hi-C links were considered unreliable and removed. After this quality filtering, 109 putative plasmid clusters made up of 379 contigs were retained for analysis (Fig. S6a). The remaining putative plasmids were classified as associating with an AMR gene if the putative plasmid was connected via Hi-C to an AMR contig at least five times (Fig. S6b).

### Transposons
To ensure we captured Hi-C associations between bacteria and plasmids only, we filtered out any remaining contigs that were identified as transposons, even if these were additionally identified as MAGs or putative plasmids. Transposons and IS elements were identified by performing a homology search with BLASTp on predicted genes from all contigs using an *e*-value < 0.01 against all known transposase proteins from the databases from IS finder[52] available from https://github.com/thanhleviet/ISfinder-sequences/blob/master/README.md and from Tn3 Transposon Finder[53] available from https://tncentral.proteininformationresource.org/TnFinder.html. Protein-coding genes were predicted from all contigs using prodigal in metagenomic mode using the option '-p meta' available from https://github.com/hyattpd/Prodigal[54].

### Analysis and reproducibility
Hi-C link counts were normalised by both MAG abundance and putative plasmid size, as these both would affect the number of links detected (Fig S5). A MAG-putative plasmid adjacency matrix was generated from the processed Hi-C linkage data, and data was handled using the packages phyloseq[62] and igraph. Any Hi-C connections represented by five or fewer links were deemed unreliable and removed. Networks were visualised using the ggnetwork[63] using graphopt layout. Network statistics for the five major host classes were generated with the *bipartite::networklevel* function[64]. Network clusters were calculated with the *igraph::fastgreedy.community* function. Phylogenetic trees and their attributes were visualised with the *ggtree* package[65]. To test for differences in the number of plasmids harboured by different bacterial classes we used a non-parametric Kruskal–Wallace test, and differences in average network degree and phylogenetic breadth between plasmids with and without AMR were tested using a non-parametric Wilcoxon test. We calculated the phylogenetic breadth of bacterial hosts by measuring the mean distance between bacterial tips using a rooted phylogenetic tree generated by Phylophlan. The mean phylogenetic distance was calculated with the *ape::branching.time* function. We also calculated maximum and median distances between branch tips and applying these metrics also demonstrated significant phylogenetic host breadth of plasmids associated with AMR genes.

### Reporting summary
Further information on research design is available in the Nature Portfolio Reporting Summary linked to this article.

## Data availability
Sequencing data are available in FASTQ format at SRA accession PRJNA506462. All data and code for both the ecological model and the Hi-C metagenome data are available at https://doi.org/10.17605/OSF.IO/K8PMF. Source data to reproduce figures is available as in the source data file. Source data are provided with this paper.

## Code availability
All data and code for both the ecological model and the Hi-C metagenome data are available at https://doi.org/10.17605/OSF.IO/K8PMF.

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

## Acknowledgements

We thank Mike Brockhurst for the discussion about the manuscript and Suzanne Kay for help with the data analysis. This research is funded by NERC, grant number NE/S000771/1. B.I.S. is supported by a Royal Commission for the Exhibition of 1851 Research Fellowship. D.S. is supported by a BBSRC Discovery Fellowship BB/X010473/1. T.S. is in part supported by the programme Understanding Antimicrobial Resistance award no. 2018-67017-27630 from the USDA National Institute of Food and Agriculture. A.N. is supported by Grant MR/N0137941/1 for the GW4 BIOMED Medical Research Council Doctoral Training Partnership, awarded to the Universities of Bath, Bristol, Cardiff, and Exeter from the Medical Research Council/UK Research and Innovation.

## Author contributions

D.S., A.B., A.R. and A.N. conceived and designed the study. A.N. conducted the modelling work. A.R., T.S. and B.I.S. analysed the data. D.S. and A.R. wrote the first manuscript draft and T.S., E.M.T., A.B., A.N. and B.I.S. contributed to the interpretation of the findings and writing the manuscript. The data and code for analysis supporting the results will be archived in an appropriate public repository and the data DOI will be included at the end of the article.

## Competing interests

The authors declare no competing interests.
