## [Peer Review File · Nature Communications]

REVIEWER COMMENTS

Reviewer #1 (Remarks to the Author):

Reviewer #2 (Remarks to the Author):

The manuscript by Risely and Newbury et al. explores how carriage of antimicrobial resistance genes affects plasmid-host networks. The authors first use a computational model to simulate plasmid-host dynamics based on community structure, conjugation rate, and fitness costs to predict that plasmids that increase fitness of even one host in a community (e.g. contain a resistance gene) spread to more hosts. The predictions of this model are then tested using previously published Hi-C data which confirms that plasmids containing AMR genes, inferred by a relatively recently-developed machine learning package, are linked to a greater number of hosts than non-AMR plasmids. Overall, the manuscript is well-written and presents an interesting story by combining modelling and metagenomic approaches. The modelling is clever and it is exciting to see how the authors support the predictions with careful re-analysis of a valuable public dataset.

Main comment/criticism:

The implication of the manuscript (coded directly in the model) is that the wastewater samples analysed constitute a community, and that AMR genes, in providing a benefit, structure the network of plasmid carriage towards generalism. There are a few issues with this. First, as the authors recognise (line 197ff) the causality could be reversed: a broad-host range infectious (parasitic) plasmid could be predisposed to neutrally acquiring and disseminating AMR genes, since transposition e.g. of AMR transposons may be triggered by conjugation. Second, the wastewater community is an aggregate of organisms pooled from different sources, and then (as I understand) 24 different wastewater samples (l. 314) were pooled for the Hi-C metagenomics. Selection and transmission of plasmids could vary amongst the input samples which might account for some of the observations. For example, if samples varied in their levels of historic antibiotic exposure then the benefits of the AMR plasmids would likewise vary and is not immediately clear what the implications would be for the model.

One way that these issues could be investigated is by comparing the overall patterns from the wastewater metagenomes, in which AMR is likely to provide (or have provided) a benefit on average, with metagenomes collected from microbiomes in which AMR genes are unlikely to be beneficial, i.e. samples that may not be frequently exposed to antibiotics to see how that affects the network of AMR-plasmids vs non-AMR-plasmids. There are several other datasets that could be used such as Yaffe and Relman (2020) that is cited in this paper (healthy human gut microbiota), or Cuscó et al. (2022) (healthy

canine faeces). However, I understand that this would be a lot more work to now add to the manuscript, and it may be sufficient for the authors to caveat their main conclusions (especially in the abstract lines 39-41).

Other comments:

- Using the machine learning-based programs to identify plasmids could lead to some false positives, which is addressed as a limitation by the authors, who also use some measures to eliminate incorrectly classified contigs. The authors also, importantly, used measures to reduce the impact of spurious links in the Hi-C data, but do address in the discussion that these false links are still a possibility.

- Line 59: delete double space "...suggest that plasmids that have..."

- Line 84-85: the placement of the citations make it seem like the papers being referenced are the ones that used Hi-C metagenomics to link mobile genetic elements to hosts in wastewater. Perhaps move them to right after "Hi-C metagenomics".

- Line 94: "degree" is an important term for understanding the rest of the paper, maybe needs more definition here to understand it - higher "degree" = linked to more taxa? So that a wider audience can understand

- Line 121: says "(Fig. 2A-C)" but figure 2 only contains A and B. Either wrong figure or text needs updating.

- Line 127: same as above, "Fig. 2B,C" needs updating.

- Line 142: rephrase "higher number of not linked sub networks". Does this mean separate sub networks that aren't connected to other sub networks?

- Line 145: talks about visualising 15 putative plasmids but figure 4 only contains 10 plasmids.

- What sensitivity analyses were performed on the model? Line 213 refers to sensitivity analysis and line 295 refers to a 'range of parameter values' but I can't see these stated anywhere.

- Line 280: spelling error ("plasmid fee")

- Line 333: does 'connected' means a Hi-C connection or genetic linkage?

- Line 336: might be good to give an explanation for "as their ubiquity can generate spurious associations" (talking about transposons), or include a citation e.g. McCallum et al. 2023 (<https://doi.org/10.1099/mgen.0.001030>) found that repetitive regions such as IS elements can result in spurious links.

- Line 336: also, replace the comma after "as their ubiquity can generate spurious associations" with a full-stop.

- Line 339: delete the hyphen after full-stop "...very few contigs (~4%).-- We then used..."

- I was also a bit confused by this paragraph — how could a MAG or contig be assigned to ‘transposon’ if “Any MAG or plasmid contig that was also identified as a transposon was removed from the dataset”? A flow chart might be useful to explain the analysis pipeline.

- Line 347: mentions Table S1 but I cannot find this table anywhere.

- Line 349: mentions genome size and completeness scores from CheckM. It might be good to include a supplementary table of all the MAGs, taxonomy, completeness, contamination etc. This may be what Table S1 is but I cannot find it in the current manuscript files.

- Line 418 refers to a t-test but line 137-138 describe a Wilcoxon test (which seems more suitable).

- Figure 1 and lines 104-110. It is still a bit unclear to me why increasing conjugation rate does not necessarily cause an increase in degree, and instead there is a peak at intermediate conjugation rates. Higher conjugation rates would both increase the total amount of plasmid and increase intra-strain transmission. From the text I understood that it was the *increase* that was greatest at intermediate conjugation rate (which makes more sense, i.e. the marginal effect of positive interactions is strongest at intermediate conjugation levels). However, the annotation and legend of Figure 1 confused me again, as it seems to be reporting just the degree across different positive interaction levels. This should be clarified.

- The legend to Figure 1 should indicate that the scale bars for degree are different between 1A and 1B.

- Line 615: as mentioned previously, the legend of Figure 2 needs to be updated as it is describing a figure with panels A-C, whereas the figure in the current manuscript only contains A and B.

- Figure 3 and Figure 4. The nature of the tree should be indicated (assuming it represent taxonomy?)

- Line 629: what does the size of the circles/stars represent? This applies to figure 2 as well.

- Line 638: as mentioned previously, the figure only contains 10 plasmids but the legend mentions 15.

- Line 678: correct spelling of “PlassClass” to “PlasClass”.

- Figure 3: Am I right that there are more plasmids without AMR genes than plasmids with? Might this neutrally affect network structure, i.e. more plasmids might increase the number of compartments, since there are just more opportunities for singleton plasmids?

- Figure 4 might benefit from having (faint) class section colours extend behind the rest of the figure to make it easier to see which class the linked MAG is in

- Figure S3 – what do the red x, y, and z mean on the figure? I think these are left over from a previous version and should be removed?

- The Rmarkdown report is clear and useful, but it doesn’t display correctly within GitHub (I had to download and open it in a local browser window).

Reviewer #3 (Remarks to the Author):

In this manuscript, Risely, Newbury et al. explore the hypothesis that beneficial plasmids are more likely to spread across bacterial communities thanks to source-sink dynamics, even if the focal plasmid is only beneficial to a single bacterial clone. To test this hypothesis, they develop a mathematical model (which is an extension of a previously published model) that shows that at intermediate conjugation rates, plasmids beneficial to a subset of hosts spread rapidly, even if deleterious to other hosts. Then, the authors move to prove the insights provided by the model in real settings and analyze a wastewater community plasmidome using Hi-C (from a previously published dataset). Their results show that, as predicted, AMR plasmids tend to be more widely distributed across bacterial species and clones than non-AMR plasmids.

These results are important, as they shed light on long-standing questions about the factors that govern plasmid dissemination using data from real-life (and not experimental) microbial communities. In general, the paper is well written, although I believe that it could be improved by adding more detailed explanations in some parts of the manuscript (see below for specific comments).

General comments:

- One thing that would definitively improve the manuscript is that the interpretation of the experimental results are based on the assumption that AMR plasmids are beneficial in the wastewater environment. While I believe this is a safe assumption to make, I think it should be better justified in the paper. For instance, it would be great to add a couple of references showing which antibiotics (and in which concentrations) are typically present in wastewaters. In addition, it would be also great to discuss the assumption of beneficial AMR in contrast to recent (and not so recent) meta-analyses that demonstrate that plasmids tend to be costly, and that the more AMR genes a plasmid carries, the more it costs (PMID: 37455716 & 25861386).
- Along this line of thought, I was a bit surprised that there's no reference to the specific AMR genes found in the plasmids. This could add valuable insights to the manuscript. For instance, how does the AMR content of each plasmid affect its degree within the network? Perhaps some supplementary analyses on putative plasmid characteristics (e.g., incompatibility group, AMR content, predicted mobility) would help to better understand plasmid distribution across the network.
- I'm not sure about the title of the article, as from the data shown, it is difficult to make specific predictions (particularly regarding network structure) beyond the idea that beneficial plasmids spread more rapidly. The authors could consider changing the title of the article to something that more accurately reflects the findings of the study.
- In my opinion, the result section starts a bit too abruptly. I think that it would benefit the reader if a succinct explanation of how the model works is provided at the beginning of the results section.
- As far as I understand, the model only deals with plasmids of the same incompatibility group (i.e. unable to coexist within the same cell). What would happen when multiple compatible plasmids compete? This is relevant because different plasmids typically coexist in real-life settings, as demonstrated by their wastewater data.

- Regarding the Hi-C results, I'm a bit worried about the fact that most AMR plasmids are associated with proteobacteria. This could bias the results. How this affects the author's interpretations? Is the subnetwork of proteobacteria AMR vs No AMR plasmids different?
- In the methods section and the R notebook, there's information about 'phylogenetic signal'. Why these analyses have been eliminated from the manuscript?
- In the abstract, the authors state that they use 'machine learning' to analyze Stalder's Hi-C results. This info is repeated again in the introduction. However, I failed to understand what's the novelty of this approach. From the methods section, it seems that the authors just used published methodologies (based on ML) to re-analyze the data. If that's the case, please remove the ML reference from the abstract, as it kind of misleadingly suggests that the authors employed a novel methodology rather than a published one.

Minor comments (mostly involving clarity):

- L80. Typo: whose
- L110: Specify the Supplementary Figure(s) you're referring to.

- Figure 1: Consider converting the continuous colour coding of plasmid degree to discrete values for clarity. Maintaining consistent scales for panels A and B will aid comparison.
- Figure 1 legend: What does 'All plasmids may or may not interact positively (increase host fitness) with a set of hosts' mean? I enjoyed the panel c description in the figure legend; it really helps understand the model's results.
- L121. Panel C is missing.
- I believe proteobacteria is no longer the correct name for that taxonomic unit. Unfortunately, we should use pseudomonadota now.
- L126. I agree that most plasmids seem to cluster with proteobacteria, but I think the specific number should be given here (X/289 putative plasmids). Similarly, the statement in L135-136 could benefit from some statistical support.
- Figure 2. While it might be seen as obvious for an expert, these figures are usually hard to understand for non-familiarized readers. Please provide a more detailed description of the network. What do nodes and links represent? What drives node and plasmid sizes? What drives the size (line width) of the links? Is it abundance?
- Figure 3A: Perhaps the network visualization can be tweaked a little bit so isolated sub-networks become more conspicuous. This will help the reader assess the modularity and the # of compartments.
- Can you please provide a small definition of what 'nested' means in a network?

- L145: In the figure, there are only 10 plasmids. I've gone through the R notebook and found the 15 plasmids the authors are referring to, but unfortunately, they're not colour coded according to whether they are AMR or not, so it's difficult to interpret.

- Figure 4. Some bacterial classes in the phylogenetic tree are not indicated. Why is that?

- While the R markdown notebook is a valuable addition to the manuscript, it seems that there's much more information there than in the manuscript. This is a bit confusing, so I suggest keeping the same structure as in the manuscript.

Reviewer #4 (Remarks to the Author):

In the manuscript "Antimicrobial resistance genes predict plasmid spread and network structure in a wastewater sample," the authors combine a mathematical model with data obtained from wastewater environments to explore the interactions between bacterial strains, plasmids, and the transmission of antibiotic resistance in microbial ecosystems. This model is designed to simulate the dynamics of plasmid-host associations, especially considering those plasmids that carry antibiotic resistance genes. Their findings, derived from the model and corroborated with empirical wastewater data, revealed that plasmids harboring resistance genes tend to form associations with a wider range of bacterial hosts. As a result, these plasmids can efficiently promote the spread of antibiotic resistance across diverse bacterial species.

The mathematical model used to examine the dynamics of host-plasmid networks within microbial communities is an extension of a previously-published work, expanded to consider multiple bacterial strains, plasmids, and resources. Central to their model's dynamics was the distinction between intra-strain and inter-strain plasmid transfers, emphasizing the potential spread of antibiotic resistance within and between bacterial populations. A simplification of the model is that a bacterial cell could carry only one type of plasmid from a specific incompatibility group. The authors used computational techniques to solve their model's equations, basing their simulations on real-world data. The results highlighted how factors such as plasmid transfer rates and the benefits offered by plasmids influenced the host-plasmid network structure.

The model provided an examination of plasmid dynamics in populations competing for limiting resources, offering valuable insights into host-plasmid associations in microbial populations. However, in real-world microbial communities, bacteria often engage in more intricate interactions, such as metabolite secretion leading to cross-feeding between species. While the model's current structure might not capture these dynamics, a discussion on their potential implications and relevance could have enriched the manuscript. How do the authors envision that diverse microbial interactions, beyond just resource competition, could influence the distribution of plasmids within these communities? Such an inclusion in the discussion would have set the findings in a broader context, helping readers understand the complexity and multifaceted nature of microbial interactions beyond the scope of the current study.

In summary, this is a well-written manuscript that addresses an important problem in plasmid biology. The combination of mathematical modeling, computer simulations and real-world data from wastewater environments provides an interesting approach that has the potential to be a powerful tool for predicting antibiotic resistance evolution and for evaluating interventions designed to control the spread of plasmid-borne antibiotic resistance.

Minor comments:

L55. what do the authors mean by “realised plasmid host range”?

L155. In the opening line: "Or model predicts..." should be "Our model predicts...".

Detailed response to REVIEWER COMMENTS

Reviewer #2 (Remarks to the Author):

The manuscript by Risely and Newbury et al. explores how carriage of antimicrobial resistance genes affects plasmid-host networks. The authors first use a computational model to simulate plasmid-host dynamics based on community structure, conjugation rate, and fitness costs to predict that plasmids that increase fitness of even one host in a community (e.g. contain a resistance gene) spread to more hosts. The predictions of this model are then tested using previously published Hi-C data which confirms that plasmids containing AMR genes, inferred by a relatively recently-developed machine learning package, are linked to a greater number of hosts than non-AMR plasmids. Overall, the manuscript is well-written and presents an interesting story by combining modelling and metagenomic approaches. The modelling is clever and it is exciting to see how the authors support the predictions with careful re-analysis of a valuable public dataset.

Main comment/criticism:

The implication of the manuscript (coded directly in the model) is that the wastewater samples analysed constitute a community, and that AMR genes, in providing a benefit, structure the network of plasmid carriage towards generalism. There are a few issues with this. First, as the authors recognise (line 197ff) the causality could be reversed: a broad-host range infectious (parasitic) plasmid could be predisposed to neutrally acquiring and disseminating AMR genes, since transposition e.g. of AMR transposons may be triggered by conjugation.

RESPONSE: Theory (here and in Newbury et al. 2021, PNAS, 119 (22), e2118361119) and experiment (see Newbury et al. 2021), where the same broad host range plasmid pkjk5 is a generalist when beneficial and a specialist when costly, point to a causal connection. However, as the reviewer suggests this effect is likely enhanced when generalist plasmids in a microbial community interact with a wider range of hosts and genetic material. There is little doubt that antibiotics increase the prevalence of plasmids that provide resistance to antibiotics in strains and at the community level. In this context, plasmids are much more likely to transmit within the community (as shown by our model) thereby changing the network structure to becoming more connected. Ecological and (co)-evolutionary dynamics, by increasing plasmid prevalence and reducing plasmid fitness costs, may then lead to long term changes where these plasmids keep their role as generalists even in the absence of selective pressure.

Second, the wastewater community is an aggregate of organisms pooled from different sources, and then (as I understand) 24 different wastewater samples (l. 314) were pooled for the Hi-C metagenomics. Selection and transmission of plasmids could vary amongst the input samples which might account for some of the observations. For example, if samples varied in their levels of historic antibiotic exposure then the benefits of the AMR plasmids would likewise vary and is not immediately clear what the implications would be for the model. One way that these issues could be investigated is by comparing the overall patterns from the wastewater metagenomes, in which AMR is likely to provide (or have provided) a benefit on average, with metagenomes collected from microbiomes in which AMR genes are unlikely to be beneficial, i.e. samples that may not be frequently exposed to antibiotics to see how that affects the network of AMR-plasmids vs non-AMR-plasmids. There are several other datasets that could be used such as Yaffe and Relman (2020) that is cited in this paper (healthy human gut microbiota), or Cuscó et al. (2022) (healthy canine faeces). However, I understand that this would be a lot more work to now add to the manuscript, and it may be sufficient for the authors to caveat their main conclusions (especially in the abstract lines 39-41).

RESPONSE: The sample constitutes a wastewater influent community (before treatment)- this is a dynamic community. Please note that sampling from multiple locations is a misunderstanding, our analysis is based on 1 sample from 1 location (Moscow WWTP in Idaho (USA)) collected over a 24

hour period. Further multiple samples and a mix from different environments of origin is not a problem for our overall assumption as average plasmid benefit is higher with an association with AMR genes in environments with varying antibiotic and biocide concentrations. We now cite several studies that report the antibiotic concentrations in influent wastewater (before treatment), with antibiotics belonging to a range of different classes commonly found at for microbial communities' relevant concentrations.

Other comments:

- Using the machine learning-based programs to identify plasmids could lead to some false positives, which is addressed as a limitation by the authors, who also use some measures to eliminate incorrectly classified contigs. The authors also, importantly, used measures to reduce the impact of spurious links in the Hi-C data, but do address in the discussion that these false links are still a possibility.

RESPONSE: Yes we agree and indeed discuss this, with future advancements in methods likely improving plasmid identification. We also decided to use a recently published pipeline (geNomad) to get more robust results.

- Line 59: delete double space "...suggest that plasmids that have..."

RESPONSE: Corrected as suggested.

- Line 84-85: the placement of the citations make it seem like the papers being referenced are the ones that used Hi-C metagenomics to link mobile genetic elements to hosts in wastewater. Perhaps move them to right after "Hi-C metagenomics".

RESPONSE: Corrected as suggested.

- Line 94: "degree" is an important term for understanding the rest of the paper, maybe needs more definition here to understand it - higher "degree" = linked to more taxa? So that a wider audience can understand

RESPONSE: Added as suggested.

- Line 121: says "(Fig. 2A-C)" but figure 2 only contains A and B. Either wrong figure or text needs updating.

RESPONSE: Corrected.

- Line 127: same as above, "Fig. 2B,C" needs updating.

RESPONSE: Corrected.

- Line 142: rephrase "higher number of not linked sub networks". Does this mean separate sub networks that aren't connected to other sub networks?

RESPONSE: We have rephrased this to "higher number of separated sub networks"

- Line 145: talks about visualising 15 putative plasmids but figure 4 only contains 10 plasmids.

RESPONSE: Corrected as suggested.

- What sensitivity analyses were performed on the model? Line 213 refers to sensitivity analysis and line 295 refers to a 'range of parameter values' but I can't see these stated anywhere.

RESPONSE: Parameter values and distributions for the population model and from which they are taken are given in the methods section from line 342.

- Line 280: spelling error ("plasmid fee")

RESPONSE: Corrected.

- Line 333: does 'connected' means a Hi-C connection or genetic linkage?

RESPONSE: Connections refer to Hi-C connections, which we assume to represent genetic linkage. We have clarified in the MS.

- Line 336: might be good to give an explanation for "as their ubiquity can generate spurious associations" (talking about transposons), or include a citation e.g. McCallum et al. 2023 (found that repetitive regions such as IS elements can result in spurious links).

RESPONSE: The spurious links referred to stem from when mobile genetic elements are incorrectly classified as MAGs (or indeed plasmids). To ensure we are capturing associations between bacteria and plasmids only, we removed all potential transposons from the dataset, even if they were also classified as MAGs, plasmids, or contigs hosting AMR genes. This reduces the risk of recoding strong MAG-plasmid associations that are in reality due to transposons. We have clarified this in the MS and cited the McCallum paper suggested.

- Line 336: also, replace the comma after "as their ubiquity can generate spurious associations" with a full-stop.

RESPONSE: Corrected.

- Line 339: delete the hyphen after full-stop "...very few contigs (~4%).-- We then used..."

RESPONSE: Corrected.

- I was also a bit confused by this paragraph — how could a MAG or contig be assigned to 'transposon' if "Any MAG or plasmid contig that was also identified as a transposon was removed from the dataset"? A flow chart might be useful to explain the analysis pipeline.

RESPONSE: Thank you for pointing out this inconsistency. All contigs identified as potential transposons by ISfinder were removed from the dataset and we have clarified this in the manuscript. We have now added a methods flowchart (Fig S5) as requested.

- Line 347: mentions Table S1 but I cannot find this table anywhere.

RESPONSE: Thank you for pointing out this oversight. This table is of the Phylophlan taxonomy results per MAG as has been uploaded as a large supplementary table. This table also includes all MAG metadata including abundance and completeness.

- Line 349: mentions genome size and completeness scores from CheckM. It might be good to include a supplementary table of all the MAGs, taxonomy, completeness, contamination etc. This may be what Table S1 is but I cannot find it in the current manuscript files.

RESPONSE: This table on MAG metadata has been now uploaded as a supplementary excel file (Table S1).

- Line 418 refers to a t-test but line 137-138 describe a Wilcoxon test (which seems more suitable).

RESPONSE: Corrected. It is indeed a Wilcoxon test as the data is highly positively skewed. A t-test or general linear model on transformed data produces the same results.

- Figure 1 and lines 104-110. It is still a bit unclear to me why increasing conjugation rate does not necessarily cause an increase in degree, and instead there is a peak at intermediate conjugation rates. Higher conjugation rates would both increase the total amount of plasmid and increase intra-strain transmission. From the text I understood that it was the *increase* that was greatest at intermediate conjugation rate (which makes more sense, i.e. the marginal effect of positive interactions is strongest at intermediate conjugation levels). However, the annotation and legend of

Figure 1 confused me again, as it seems to be reporting just the degree across different positive interaction levels. This should be clarified.

RESPONSE: Figure 1 and accompanying legend have been modified to explain this point.

- The legend to Figure 1 should indicate that the scale bars for degree are different between 1A and 1B.

RESPONSE: For simplicity, Figure 1 has been edited to focus more specifically on the main message of the manuscript: the effect of positive vs negative interactions on network structure. As such we now only display a single heatmap which shows how the degree of a plasmid changes with its number of positive interactions, whilst both other plasmids exhibit only negative interactions.

- Line 615: as mentioned previously, the legend of Figure 2 needs to be updated as it is describing a figure with panels A-C, whereas the figure in the current manuscript only contains A and B.

RESPONSE: Corrected.

- Figure 3 and Figure 4. The nature of the tree should be indicated (assuming it represent taxonomy?)

RESPONSE: This is correct. We have clarified this in the figure legends.

- Line 629: what does the size of the circles/stars represent? This applies to figure 2 as well.

RESPONSE: This information is presented in the figure legend, both visually and in the text. Circles represent bacteria, stars represent plasmids.

- Line 638: as mentioned previously, the figure only contains 10 plasmids but the legend mentions 15.

RESPONSE: Corrected.

- Line 678: correct spelling of "PlassClass" to "PlasClass".

Response: We decided to use geNomad which is a new pipeline, see below.

- Figure 3: Am I right that there are more plasmids without AMR genes than plasmids with? Might this neutrally affect network structure, i.e. more plasmids might increase the number of compartments, since there are just more opportunities for singleton plasmids?

RESPONSE: In our new analysis using the new pipeline geNomad (Camargo et al. "Identification of mobile genetic elements with geNomad." *Nature Biotechnology* 2023), the subnetworks that compare plasmid networks with and without AMR are much more equal in size. This suggests that any differences in network structure are not a result of network size. Moreover, we have also now included a comparison of Proteobacteria subnetworks and found similar differences in these networks compared to the full network.

- Figure 4 might benefit from having (faint) class section colours extend behind the rest of the figure to make it easier to see which class the linked MAG is in

RESPONSE: We have added these to Figure 4.

- Figure S3 – what do the red x, y, and z mean on the figure? I think these are left over from a previous version and should be removed?

RESPONSE: These were originally linked to plasmid clusters in the main figures. Thanks for pointing this out – they have been removed.

- The Rmarkdown report is clear and useful, but it doesn't display correctly within GitHub (I had to download and open it in a local browser window).

RESPONSE: Unfortunately this is normal for github – it doesn't open html files without them being downloaded first. We are unsure what the solution to this is other than additionally include a PDF which opens within GitHub. The R markdown file has been updated and we have additionally added a Hi-C processing workflow diagram.

Reviewer #3 (Remarks to the Author):

In this manuscript, Risely, Newbury et al. explore the hypothesis that beneficial plasmids are more likely to spread across bacterial communities thanks to source-sink dynamics, even if the focal plasmid is only beneficial to a single bacterial clone. To test this hypothesis, they develop a mathematical model (which is an extension of a previously published model) that shows that at intermediate conjugation rates, plasmids beneficial to a subset of hosts spread rapidly, even if deleterious to other hosts. Then, the authors move to prove the insights provided by the model in real settings and analyze a wastewater community plasmidome using Hi-C (from a previously published dataset). Their results show that, as predicted, AMR plasmids tend to be more widely distributed across bacterial species and clones than non-AMR plasmids.

These results are important, as they shed light on long-standing questions about the factors that govern plasmid dissemination using data from real-life (and not experimental) microbial communities. In general, the paper is well written, although I believe that it could be improved by adding more detailed explanations in some parts of the manuscript (see below for specific comments).

General comments:

- One thing that would definitely improve the manuscript is that the interpretation of the experimental results are based on the assumption that AMR plasmids are beneficial in the wastewater environment. While I believe this is a safe assumption to make, I think it should be better justified in the paper. For instance, it would be great to add a couple of references showing which antibiotics (and in which concentrations) are typically present in wastewaters. In addition, it would be also great to discuss the assumption of beneficial AMR in contrast to recent (and not so recent) meta-analyses that demonstrate that plasmids tend to be costly, and that the more AMR genes a plasmid carries, the more it costs (PMID: 37455716 & 25861386).

RESPONSE: We have now cited a few studies in the introduction that have analysed antibiotic concentrations in wastewater. See text lines 95-97.

Thank you for the suggested references. They have been included in the discussion. See text lines 234-238: "Two meta-analyses observed that the accumulation of AMR resulting from mutations of chromosomal genes entails much stronger fitness cost than the accumulation of transferable AMR genes from plasmids^{40,42}. This phenomenon may contribute to the observed dominance of transferable AMR genes in the current multidrug resistance epidemic in enterobacteria."

- Along this line of thought, I was a bit surprised that there's no reference to the specific AMR genes found in the plasmids. This could add valuable insights to the manuscript. For instance, how does the AMR content of each plasmid affect its degree within the network? Perhaps some supplementary analyses on putative plasmid characteristics (e.g., incompatibility group, AMR content, predicted mobility) would help to better understand plasmid distribution across the network.

RESPONSE: We have now added information on the AMR genes found on the AMR-plasmids, and included an additional supplementary figure (Fig. S3) that highlights how the AMR genes cluster alongside bacterial hosts.

- I'm not sure about the title of the article, as from the data shown, it is difficult to make specific predictions (particularly regarding network structure) beyond the idea that beneficial plasmids

spread more rapidly. The authors could consider changing the title of the article to something that more accurately reflects the findings of the study.

RESPONSE: We have changed the title.

- In my opinion, the result section starts a bit too abruptly. I think that it would benefit the reader if a succinct explanation of how the model works is provided at the beginning of the results section.

RESPONSE: An additional paragraph has been added to the beginning of the result section.

- As far as I understand, the model only deals with plasmids of the same incompatibility group (i.e. unable to coexist within the same cell). What would happen when multiple compatible plasmids compete? This is relevant because different plasmids typically coexist in real-life settings, as demonstrated by their wastewater data.

RESPONSE: We now discuss this in the text. See lines 261-275: “The modelling approach employed here is more complicated than most plasmid population models, due to the incorporation of multiple hosts, plasmids and resources. However, some simplifying assumptions were necessary. Firstly, we assume bacterial hosts interact only through competition for resources. While in reality bacteria interact via diverse mechanisms such as the excretion of both nutrients and toxins, such features are unconnected to the core focus of the present works – the increased spread of plasmids throughout a host community due to benefits conferred within populations. That said, the impact of host community structure (as a result of inter and intra-specific interactions) on host-plasmid networks is itself an interesting question for future research. The model was also simplified by considering multiple plasmids from within the same incompatibility group. This avoids making further assumptions about how the fitness effects of multiple plasmids interact in natural communities. Furthermore, it has been shown previously that a plasmid that is not competing with other plasmids for hosts will form a better connected plasmid-host network when it is beneficial 9. Thus, it is not expected that considering multiple plasmids which are compatible with each other would change our overall result. Though again, the exact effects of the distribution of incompatibility groups on plasmid-hosts networks is itself a useful line of inquiry for future research.”

- Regarding the Hi-C results, I’m a bit worried about the fact that most AMR plasmids are associated with proteobacteria. This could bias the results. How this affects the author's interpretations? Its the subnetwork of proteobacteria AMR vs No AMR plasmids different?

RESPONSE: We have added Proteobacteria (now Pseudomonadota) subnetworks to the manuscript (Fig. S4), the network structures found resemble that of the overall analysis.

- In the methods section and the R notebook, there’s information about ‘phylogenetic signal’. Why these analyses have been eliminated from the manuscript?

RESPONSE: In an earlier version of this manuscript, we included an analysis on phylogenetic signal of plasmids. However, we felt that the number of results was diluting the overall message and clarity, and it was not directly relevant to the major question tackled in this paper (are plasmids with AMR genes disproportionately important for connecting bacterial networks?). We therefore removed this analysis.

- In the abstract, the authors state that they use ‘machine learning’ to analyze Stalder’s Hi-C results. This info is repeated again in the introduction. However, I failed to understand what’s the novelty of this approach. From the methods section, it seems that the authors just used published methodologies (based on ML) to re-analyze the data. If that’s the case, please remove the ML reference from the abstract, as it kind of misleadingly suggests that the authors employed a novel methodology rather than a published one.

RESPONSE: We have removed the word ‘novel’ from the discussion in relation to our use of machine learning methods. However, we don’t claim that our machine learning methods are novel in the

abstract or the introduction. We mention it because it is an important part of our methods, and we believe that methodological approaches should be clear both in the abstract and the introduction, novel or otherwise. To remove it from the abstract could mislead readers to think that we are identifying plasmids based on other methods.

Minor comments (mostly involving clarity):

- L80. Typo: whose

- L110: Specify the Supplementary Figure(s) you're referring to.

RESPONSE: Corrected.

- Figure 1: Consider converting the continuous colour coding of plasmid degree to discrete values for clarity. Maintaining consistent scales for panels A and B will aid comparison.

RESPONSE: For simplicity, Figure 1 has been edited to focus more specifically on the main message of the manuscript: the effect of positive vs negative interactions on network structure. As such we now only display a single heatmap which shows how the degree of a plasmid changes with its number of positive interactions, whilst both other plasmids exhibit only negative interactions. A continuous scale is used because each tile represents a mean value derived from many simulations.

- Figure 1 legend: What does 'All plasmids may or may not interact positively (increase host fitness) with a set of hosts' mean? I enjoyed the panel c description in the figure legend; it really helps understand the model's results.

RESPONSE: This text has been amended in the manuscript.

- L121. Panel C is missing.

RESPONSE: We are not entirely sure what this refers to, but we have checked all figure references in the text.

- I believe proteobacteria is no longer the correct name for that taxonomic unit. Unfortunately, we should use pseudomonadota now.

RESPONSE: Corrected as suggested.

- L126. I agree that most plasmids seem to cluster with proteobacteria, but I think the specific number should be given here (X/289 putative plasmids). Similarly, the statement in L135-136 could benefit from some statistical support.

RESPONSE: We have added statistical support for our claim that the number of putative plasmids associated with bacterial hosts significantly differs across bacterial classes. We have also quantified that most AMR putative plasmids (21/32) cluster with Gammaproteobacteria – largely represented by the genus Acinetobacteria. We have added a supplementary figure that specifically compares the networks for proteobacteria.

- Figure 2. While it might be seen as obvious for an expert, these figures are usually hard to understand for non-familiarized readers. Please provide a more detailed description of the network. What do nodes and links represent? What drives node and plasmid sizes? What drives the size (line width) of the links? Is it abundance?

RESPONSE: We have added more details to the figure legends.

- Figure 3A: Perhaps the network visualization can be tweaked a little bit so isolated sub-networks become more conspicuous. This will help the reader assess the modularity and the # of compartments.

RESPONSE: This is tricky as the networks are highly connected – and this is reflected in the figures. We have tried to play with the network algorithm to separate out the clusters a little more, but we

feel that manually editing the network could potentially be misleading as nodes that are close to together genuinely are more strongly connected than those that are far apart.

- Can you please provide a small definition of what 'nested' means in a network?

RESPONSE: We have added the explanation: "the occurrence of generalist plasmids linking the network, with rare specialist plasmid links are already provided by the generalists"

- L145: In the figure, there are only 10 plasmids. I've gone through the R notebook and found the 15 plasmids the authors are referring to, but unfortunately, they're not colour coded according to whether they are AMR or not, so it's difficult to interpret.

RESPONSE: We have corrected this.

- Figure 4. Some bacterial classes in the phylogenetic tree are not indicated. Why is that?

RESPONSE: We have only highlighted the most common bacterial classes. Given the vast number of bacterial classes in nature, it is not possible to highlight them all. Many classes only include a small number of MAGs. We have also tried to keep the colour scheme consistent across all figures for clarity.

- While the R markdown notebook is a valuable addition to the manuscript, it seems that there's much more information there than in the manuscript. This is a bit confusing, so I suggest keeping the same structure as in the manuscript.

RESPONSE: We have edited the Rmarkdown to reflect the results.

Reviewer #4 (Remarks to the Author):

In the manuscript "Antimicrobial resistance genes predict plasmid spread and network structure in a wastewater sample," the authors combine a mathematical model with data obtained from wastewater environments to explore the interactions between bacterial strains, plasmids, and the transmission of antibiotic resistance in microbial ecosystems. This model is designed to simulate the dynamics of plasmid-host associations, especially considering those plasmids that carry antibiotic resistance genes. Their findings, derived from the model and corroborated with empirical wastewater data, revealed that plasmids harboring resistance genes tend to form associations with a wider range of bacterial hosts. As a result, these plasmids can efficiently promote the spread of antibiotic resistance across diverse bacterial species.

The mathematical model used to examine the dynamics of host-plasmid networks within microbial communities is an extension of a previously-published work, expanded to consider multiple bacterial strains, plasmids, and resources. Central to their model's dynamics was the distinction between intra-strain and inter-strain plasmid transfers, emphasizing the potential spread of antibiotic resistance within and between bacterial populations. A simplification of the model is that a bacterial cell could carry only one type of plasmid from a specific incompatibility group. The authors used computational techniques to solve their model's equations, basing their simulations on real-world data. The results highlighted how factors such as plasmid transfer rates and the benefits offered by plasmids influenced the host-plasmid network structure.

The model provided an examination of plasmid dynamics in populations competing for limiting resources, offering valuable insights into host-plasmid associations in microbial populations. However, in real-world microbial communities, bacteria often engage in more intricate interactions, such as metabolite secretion leading to cross-feeding between species. While the model's current structure might not capture these dynamics, a discussion on their potential implications and relevance could have enriched the manuscript. How do the authors envision that diverse microbial interactions, beyond just resource competition, could influence the distribution of plasmids within these communities? Such an inclusion in the discussion would have set the findings in a broader

context, helping readers understand the complexity and multifaceted nature of microbial interactions beyond the scope of the current study.

RESPONSE: This topic is now raised in the discussion section, please see lines 261-275.

In summary, this is a well-written manuscript that addresses an important problem in plasmid biology. The combination of mathematical modeling, computer simulations and real-world data from wastewater environments provides an interesting approach that has the potential to be a powerful tool for predicting antibiotic resistance evolution and for evaluating interventions designed to control the spread of plasmid-borne antibiotic resistance.

RESPONSE: Thank you.

Minor comments:

L55. what do the authors mean by “realised plasmid host range”?

RESPONSE: This means observed host range within a specific community. We have added this information to the text.

L155. In the opening line: "Or model predicts..." should be "Our model predicts..."

RESPONSE: Corrected.

REVIEWERS' COMMENTS

Reviewer #1 (Remarks to the Author):

Reviewer #2 (Remarks to the Author):

We are satisfied with the changes made to the manuscript.

Some remaining minor comments:

- Line 420 states that plasmids are likely to be >10kb in size. What is the rationale for this? Papers describing plasmid databases show a large number of plasmids <10kb, e.g. Shintani et al. 2015 doi: 10.3389/fmicb.2015.00242, Galata et al. 2019 doi: 10.1093/nar/gky1050 . Could this approach bias against small plasmids?

- Line 425. Reformat reference so it is clear it is not an exponent.

- Line 257. Typo ("Pseudomonadota").

Reviewer #3 (Remarks to the Author):

I am pleased to report that the authors have considered all of my comments and suggestions and made the necessary revisions to the manuscript. The changes made have significantly improved the quality of the paper, and I am confident that it will make a valuable contribution to the literature.

Overall, I believe this manuscript will be of great interest to researchers in Plasmid biology and AMR evolution and will serve as an important reference for future studies. I would like to congratulate the authors for their valuable work.

Detailed response to REVIEWER COMMENTS

Reviewer #1 (Remarks to the Author):

Reviewer #2 (Remarks to the Author):

We are satisfied with the changes made to the manuscript.

Some remaining minor comments:

- Line 420 states that plasmids are likely to be >10kb in size. What is the rationale for this? Papers describing plasmid databases show a large number of plasmids <10kb, e.g. Shintani et al. 2015 doi: 10.3389/fmicb.2015.00242, Galata et al. 2019 doi: 10.1093/nar/gky1050 . Could this approach bias against small plasmids?

Response: This step was necessary to increase confidence in plasmid identification. Given the current methods available, we think the error and uncertainty would be unreasonably high if we had included the short reads and importantly the overall results do not change with our new analyse (as compared to using Plasclass as a pipeline in an earlier version of the manuscript). We used the plasmid length information to judge whether the plasmid clusters were likely to represent real plasmids (based on size). The new approach may miss small plasmids, but it will not exclude conjugative or mobilizable plasmids, that are the focus of this study (see size distribution for plasmids in ref below). Importantly, it discarded lone contigs that could have been wrong plasmid predictions. Therefore overall, this approach should increase the robustness of our results.

We edited that part in the methods (line 402): “To identify contigs of plasmid origin, all contigs were run through geNomad¹⁷ using the *end-to-end* command. Because the contigs identified as plasmids were mostly constituted by short contigs (the median length was 3703 bp), we reasoned that for a contig to belong to a plasmid it should be consistently connected to at least one other such contig. To account for this, we retained only plasmid contigs that were linked to other plasmid contigs at least 15 times (n = 379 contigs). This may exclude small non-transferable plasmids, but not conjugative or mobilizable plasmids⁵⁸ that are the focus of this study.”

Ref 58: C. Smillie, M. P. Garcillán-Barcia, M. V. Francia, E. P. C. Rocha, F. d. I. Cruz, *Microbiology and Molecular Biology Reviews* 74, 434-452 (2010).

- Line 425. Reformat reference so it is clear it is not an exponent.

- Line 257. Typo (“Pseudomonadota”).

Response: Corrected as suggested.

Reviewer #3 (Remarks to the Author):

I am pleased to report that the authors have considered all of my comments and suggestions and made the necessary revisions to the manuscript. The changes made have significantly improved the quality of the paper, and I am confident that it will make a valuable contribution to the literature.

Overall, I believe this manuscript will be of great interest to researchers in Plasmid biology and AMR evolution and will serve as an important reference for future studies. I would like to congratulate the authors for their valuable work.